# Interactive Data Harmonization with LLM Agents: Opportunities and Challenges

Aécio Santos[1], Eduardo H. M. Pena[2], Roque Lopez[1], Juliana Freire[1]
[1]New York University
{aecio.santos,rlopez,juliana.freire}@nyu.edu
[2]Federal University of Technology - Paraná
eduardopena@utfpr.edu.br

## ABSTRACT

Data harmonization is an essential task that entails integrating datasets from diverse sources. Despite years of research in this area, it remains a time-consuming and challenging task due to schema mismatches, varying terminologies, and differences in data collection methodologies. This paper presents the case for agentic data harmonization as a means to both empower experts to harmonize their data and to streamline the process. We introduce Harmonia, a system that combines LLM-based reasoning, an interactive user interface, and a library of data harmonization primitives to automate the synthesis of data harmonization pipelines. We demonstrate Harmonia in a clinical data harmonization scenario, where it helps to interactively create reusable pipelines that map datasets to a standard format. Finally, we discuss challenges and open problems, and suggest research directions for advancing our vision.

**ACM Reference Format:**
Aécio Santos, Eduardo H. M. Pena, Roque Lopez, Juliana Freire. 2025. Interactive Data Harmonization with LLM Agents: Opportunities and Challenges . In *Novel Optimizations for Visionary AI Systems (NOVAS '25), June 22–27, 2025, Berlin, Germany*. ACM, New York, NY, USA, 8 pages. https://doi.org/10.1145/3735079.3735324

## 1 INTRODUCTION

Extracting insights from multiple data sources remains an arduous and time-consuming task. In fields such as biomedicine, collecting patient data requires expensive trials that typically involve only a few dozen to hundreds of patients [43]. Since this usually leads to tables with many attributes but few samples, researchers need to combine different cohorts to obtain larger sample sizes. However, since data is often collected using differing methods, combining them into compatible and comparable datasets often becomes a challenge [12]. Consider these examples from cancer research.

**Example 1** (Clinical Data Harmonization). To obtain a larger dataset for a study on endometrial cancer, researchers aim to combine samples from two patient cohorts collected independently in two studies [17, 18]. These studies produced two tables containing clinical data, which we refer to as $T_1$ and $T_2$. Each row

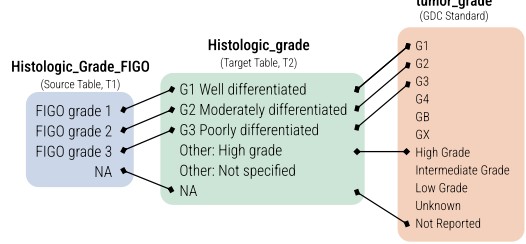

**Figure 1: Domain of attributes in different data sources.**

in each table represents an individual patient sample. Tables $T_1$ and $T_2$, contain 179 and 213 columns and 153 and 190 rows, respectively. The goal is to combine the data to obtain a table with 392 rows. However, even though these datasets were produced by the same research consortium, their schemas and naming standards differ significantly. The first challenge is identifying which columns are semantically equivalent. Once all pairs of equivalent columns have been identified, their values must be standardized. Figure 1 shows an example of a pair of equivalent attributes from $T_1$ and $T_2$ – Histologic_grade and Histologic_Grade_FIGO. The values of these attributes are represented using different terminology. To produce a harmonized table, the researchers must reconcile these values before merging the rows. For instance, they may decide that the final harmonized table $T_{target}$ will contain an attribute named histologic_grade and that the format of the value used will be from Histologic_grade, and thus we would need to map the values of $T_1$ to their corresponding values from $T_2$ (e.g., "FIGO grade 1" →"G1 Well differentiated", "FIGO grade 2" → "G2 Moderately differentiated", and so on). Note that different mappings can be applied to different attribute pairs.

**Example 2** (Harmonizing Data to a Standard Vocabulary). In a subsequent effort to foster data reuse and enable research in pan-cancer analysis [43], researchers decided to combine 10 tables containing cohorts of a variety of cancer types [7, 14, 18, 23, 28, 39, 49, 57, 64, 67]. They mapped all tables to the Genomic Data Commons (GDC) standard [31], which contains attributes commonly used in cancer research. Figure 1 shows how the values for Histologic_grade and Histologic_Grade_FIGO map to the corresponding GDC attribute tumor_grade. Once again, the acceptable values in the GDC vocabulary differ from the values used in the tables of Example 1.

**The Case for Agentic Data Harmonization.** Data harmonization involves several data integration tasks. Practitioners use spreadsheet software, bespoke scripts, and significant manual work to harmonize data [12]. These custom scripts are often not published with the data and publications, creating barriers to the reproducibility and replicability of experiments [50].

Large language models (LLMs) open new opportunities to improve data harmonization. They can answer questions about terminology, methodologies and generate code without training data. Recently, LLMs have shown promising results in data integration tasks, including column type annotation, schema matching, and entity linkage [22, 34, 44, 51, 63]. Prompting today's frontier LLMs with a question such as "*What does FIGO grade mean?*" reveals that they do have general knowledge about many topics, including biomedical research (the focus of our examples). This suggests that this information can be used in data harmonization tasks.

LLMs are becoming essential components for intelligent agents across various applications due to their capabilities in language understanding, tool utilization, and adapting to new information, resembling human intelligence and reasoning [66, 72, 73]. These capabilities have been surfaced by advancements in LLM prompting techniques that elicit reasoning, such as Chain-of-Thought [70] and ReAct [73]. These enable agents to handle complex tasks such as table understanding tasks through structured data manipulation [69], and generate reasoning traces and task-specific actions that are interleaved to complete tasks [73]. Moreover, frameworks for building agentic systems are now widely available, such as LangChain [40], Camel [41], and AutoGen [71].

This paper presents *our vision of intelligent agents that can interact with the users and data integration algorithms to synthesize data harmonization pipelines*. Such agentic systems accept user task requests and prompt the user with questions required to complete a task (e.g., to request additional context or to disambiguate the input). Similar to AutoML systems, which generate end-to-end machine learning pipelines [6, 46, 58], a data harmonization agent can produce a data processing pipeline that takes as input user data and outputs a harmonized table that satisfies the user requirements. These pipelines could then be published along with research data to document the data generation process for reproducibility.

**Agentic Data Harmonization Challenges.** Building agentic systems presents several challenges. First, harmonization scenarios are inherently complex, often involving difficult tasks such as schema matching, entity linkage, and data cleaning. These tasks require specialized methods to achieve high-quality results, and the scalability of these methods is critical for harmonizing large datasets. While general-purpose LLMs offer broad capabilities, they lack transparency [30] and are not optimized for these tasks, leading to inconsistent outputs, high computational costs, and performance bottlenecks, especially when handling large-scale datasets [21, 26].

Second, integrating algorithms for different tasks and generating cohesive pipelines is non-trivial. Unlike in AutoML systems, data harmonization pipeline synthesis cannot be driven by search algorithms that optimize well-defined evaluation metrics (e.g., model accuracy) since the quality of data harmonization pipelines cannot be as easily measured using one metric. Instead, the synthesis may need to be guided by the users, since accuracy may depend on external knowledge. For example, in the case of Fig. 1, deciding if the correct match for 'NA' is 'Not Reported' or 'Unknown' may depend on the data collection methodology. The agent should automate the laborious tasks without sacrificing accuracy and without becoming a burden: it must learn to ask questions only when necessary to avoid overwhelming the user. To add to these challenges, LLMs are brittle: (1) they often make mistakes (also known as hallucinations)

that must be identified and corrected by users; and (2) they are known to be sensitive to the prompts (i.e., small prompt changes may lead to different results) [3, 35].

**Contributions.** In this paper we discuss the opportunities and challenges for the use of agents for data harmonization. We also present `Harmonia`, a proof-of-concept prototype of an agentic data harmonization system that implements part of our vision.

`Harmonia` leverages LLM capabilities to interact with users, orchestrate specialized data integration primitives, and generate custom code when existing primitives are insufficient. These primitives implement efficient and well-established data integration algorithms that can be combined to make harmonization pipelines guided based on user feedback. Since algorithms can make mistakes, we use LLMs to evaluate the outputs of these primitives. When outputs are incorrect, the agent may take additional steps to automatically correct the errors or seek assistance from the user. We also describe how `Harmonia` can be applied in a real-world use case, highlighting the potential and limitations of its current design and implementation. To address these limitations, we identify open problems in this field and propose future research directions, building on previous work in machine learning and data management.

Our contributions can be summarized as follows: (1) We present a vision for agentic systems that help users create data harmonization pipelines by combining LLM-based reasoning, interactive user interfaces, and data integration primitives. (2) We introduce `Harmonia`, a prototype data harmonization agent that integrates `bdi-kit` [4], a library of data integration algorithms, to efficiently construct harmonization pipelines. When existing functions are insufficient, `Harmonia` leverages the LLM to dynamically generate custom code. (3) We demonstrate a real-world use case where `Harmonia` is applied to map clinical data to the GDC standard [31], addressing common issues such as differing terminology and schema. (4) We discuss challenges in designing data harmonization agents and propose a research agenda with steps to address open problems.

## 2 PRELIMINARIES

Broadly speaking, data harmonization refers to the practice of combining and reconciling different datasets to maximize their comparability or compatibility [12]. Although data harmonization goals can vary greatly depending on the data modalities and goals involved, this paper focuses on the following problem:

**Definition 1** (Tabular Data Harmonization). Given a set of source tables $T_1$, $T_2$, ..., $T_n$ and a target schema $S_{target}$ composed of a set of attributes, each attribute specifying its acceptable values, i.e., a domain, the goal is to derive a computational pipeline $\mathcal{P}$ that takes as input the source tables and applies transformation functions to values and combines tables (e.g., using union and join operations) to generate an output table $T_{target} = \mathcal{P}(T_1, T_2, ..., T_n)$ that adheres to the given target schema $S_{target}$.

Note that instead of aiming at maximizing an objective measure of comparability or compatibility between input and target, we assume the existence of the canonical data representation $S_{target}$. As seen in Examples 1 and 2, this specification $S_{target}$ consists of a set of attribute names and domain specifications that can either come from a standard data vocabulary (such as the GDC) or could be derived from the existing data in the source tables.

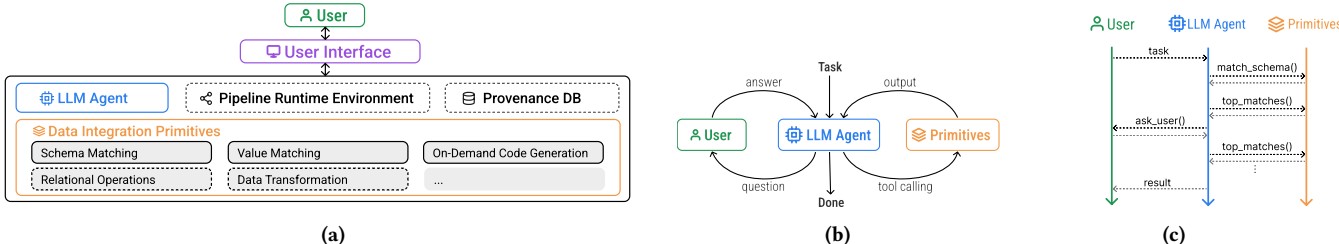

**Figure 2: (a) Components of an interactive agentic data harmonization system. Solid lines represent components implemented in `Harmonia` (see Section 4) while dashed lines represent components not yet implemented. (b) The LLM agent loop: given a task, the LLM will repeatedly execute actions (call tools or ask questions to the user) until the task is completed. (c) A sequence diagram illustrating an example of the communication workflow between the user, agent, and primitives.**

Data harmonization pipelines can be of multiple forms. A simple example is a linear sequence of data processing operations (e.g., as in scikit-learn pipelines [59] or Jupyter Notebooks [33]). However, they can also be declaratively represented as direct acyclic graphs (DAG), such as in AutoML systems [46, 58]. In practice, data harmonization pipelines are typically complex, custom scripts created iteratively through a trial-and-error process [12]. It is challenging to code and manually explore the space of pipelines to identify the most effective one for a given task. For subject matter experts without programming experience, creating these pipelines is simply out of reach. We envision that LLM-based agents can help address these challenges as we elaborate next.

## 3 AGENTIC DATA HARMONIZATION

We propose using LLM-based agents to facilitate the interactive construction of harmonization pipelines through natural language and visual interfaces. We aim to simplify the harmonization process and empower domain experts to harmonize their data effectively.

Our approach has three main components: *harmonization primitives*, *harmonization agents*, and *human-agent interaction* (Figure 2a). The system supports a two-way interaction between *users* and *agent*. While users drive the system and define the tasks to be performed, the agent aims to automate the harmonization tasks, leaving to users only decisions that need external context. By recording user-agent interactions and the derived computational pipelines, we maintain provenance of the harmonization process, supporting transparency and reproducibility and making it possible to publish the harmonized data with the pipeline used to derive it.

**Harmonization Primitives.** The bottom of Figure 2a shows a library of components that we refer to as *data integration primitives*. These are algorithms or routines that solve well-defined data integration tasks such as schema matching and value mapping. Some of these are lower-level routines that support other higher-level tasks, e.g., column-type annotation may be a component of schema-matching algorithms [22], while entity resolution and deduplication may be key to performing data standardization [13]. The set of primitives can be heterogeneous and evolve to support the system capabilities. An important requirement is that they need to be *composable* and invoked by both user and AI agents.

*Composability* is the ability to combine different primitives to create a data harmonization pipeline. For instance, output from schema matching primitives, which generates source-target attribute pairs, can feed into value mapping primitives that find equivalences between the values of those source and target attributes. Finally, the

output of value mapping primitives can be used to create a *harmonization specification* that describes the transformation of source tables $T_i$ into a target output table $T_{target}$.

**Data Harmonization Agents.** An agent senses its environment and acts upon it [55]. In our context, we define a *data harmonization agent* as a program that interacts with its environment (e.g., data, primitives, users) to autonomously complete harmonization tasks. Agents can operate at various levels of autonomy [52], from offering minimal user assistance to fully independent harmonization. In Section 4, we focus on interactive agents that respond to user queries, though more autonomy is possible within our framework.

To fulfill user requests, the agent must first decompose the problem into a series of actions of various types, including (1) the execution of existing integration primitives (*tool calling*) or (2) code generated on demand. To decompose the problem and decide on what action to take, the harmonization agent leverages an LLM, which is given descriptions of the task and integration primitives available for use. The LLM then repeatedly returns actions (e.g., a code snippet) that can be executed in a runtime environment (e.g., a Python kernel). Action outputs are fed back into the LLM, which decides if additional actions are needed. As illustrated in Figure 2b, this loop executes until the task is deemed complete.

The main loop is orchestrated by a driver code that takes inputs from the user (i.e., prompts) that describe the task to be performed. This driver is also responsible for (1) communicating with users to request inputs (e.g., when the LLM asks for task clarification or user preferences) and (2) managing the state (memory) of the agent. For example, it can also track and store the history of actions and user interactions in a *Provenance DB*. This data can support decisions about future actions, and transparency, and be used to generate harmonization *specifications* or scripts to reproduce the results.

Given the complexity of harmonization tasks, it is crucial to have high-level primitives available as building blocks for the pipeline. This allows encoding prior knowledge and using efficient algorithms known to be effective for a specific task. These primitives encompass algorithms for tasks such as schema matching, entity resolution, and value mapping. Of course, primitives can go beyond hard-coded functions that implement deterministic algorithms. For instance, they can be workflows that use LLMs to perform specific tasks such as in [44] or they could generate code on demand (e.g., to extract data from or to transform attribute values [10]).

**Human-Agent Interaction.** A key aspect of our architecture is the interaction between users and agents. In Section 4, we show

that our current prototype is proficient in text-based conversational interactions: it can parse users' input and act on them. However, to make systems more effective, we argue that future systems need to support graphical user interfaces that help users (1) to reason about the input data and the agent's output, and (2) to refine the task definition and pipelines derived by the agent. For example, as done in interactive AutoML tools [56], the system could guide the user through the harmonization process, recommend the available actions, track progress, and provide data visualizations to help the user better make sense of the data. This may help prevent common issues in natural language such as ambiguity [19, 74].

In complex tasks like harmonization, user interaction is crucial because many decisions—such as whether two terms represent the same concept—can be difficult even for domain experts. These judgments often depend on context not captured in the data, such as how it was collected or the purpose of the harmonization. Furthermore, as in any automated data integration system, LLM-based agents can make mistakes. When combined with well-designed user interfaces, users are better equipped to guide the agent, providing guardrails and safety mechanisms that neither users nor interfaces could achieve as effectively on their own.

## 4 SYSTEM PROTOTYPE & USE CASE

We implemented a proof-of-concept of our vision (Figure 2a). In what follows, we illustrate how it works with a concrete example.

**Data Integration Primitives.** We used data harmonization primitives from bdi-kit [4], an open-source Python library that we designed with the explicit goal of composability. Currently, it includes implementations of multiple *schema matching* and *value mapping* algorithms using a composable API. It also includes several classic algorithms from Koutras et al. [38] and language model-based algorithms [44, 45]. Most functions take as input a source parameter that represents the user's input DataFrame and returns the output formatted as another DataFrame. The target parameter can either be a string representing a target standard schema (e.g., 'gdc') or a target DataFrame, this allows switching between the two tasks described in Examples 1 and 2. Figure 3 shows a sample of functions integrated in Harmonia. The source code is on GitHub [24].

**LLM Agent & User Interface.** Implementing an agent entails writing carefully crafted function and task descriptions that are combined to assemble system prompts fed to the LLMs. For our prototype, we implemented tool wrappers for each of the bdi-kit functions, along with descriptions of when each should be used. We also provide descriptions of the data harmonization steps, when the LLM should request help from the user, and output formatting instructions. These descriptions are available in the code [24].

```
match_schema(source, target, method, ...)
Maps the schema of a source table to a target schema (table or predefined standard
like gdc) using a specified method.

top_matches(source, column, target, top_k, method, ...)
Finds the top-k matches between a source column and columns of a target schema.

match_values(source, target, column_mapping, method, ...)
Matches values between columns of a source and a target using a specified method,
returning one or more result tables.

materialize_mapping(input_table, mapping_spec) → DataFrame
Transforms a source table into a new table using a mapping specification.
```

**Figure 3: Some bdi-kit functions integrated in Harmonia.**

To implement tool calling, we used the Archytas [2], an open-source library for building AI agents based on the ReAct framework [73], and used the GPT-4o model (i.e., no fine-tuning is involved). Instead of using a fixed pipeline that invokes the LLM, we implement a dynamic agentic loop, as described in Figure 2b. The LLM has the flexibility to break down the task into multiple tool call instructions needed to complete the task. The user interface is implemented using Beaker [5] and is currently limited to a chat-based interface and workflow that resembles a Jupyter notebook. The system renders markdown-formatted outputs in the browser, displays the reasoning steps taken by the agent, and collects user inputs using text input boxes and feeds them back to the agent. A video demonstrating these interactions is available online [25].

**Use Case: Harmonizing a Dataset with the GDC Schema.** To demonstrate the ability to produce good data harmonization plans, we present a use case that harmonizes a dataset from Dou et al. [18] (described in Example 1) with the GDC standard. Below, we show message exchanges between the user (👤) and the agent (🤖), along with explanations of the actions taken by the agent to answer user queries. As the interactions show, Harmonia chooses proper tools to solve the task, which includes generating and executing Python code, as well as bdi-kit functions with the appropriate parameters.

The user starts by requesting the system to load a CSV file that contains the attributes of interest:

> 👤 Load the file dou.csv as a dataframe and subset it to the following columns: Country, Histologic_Grade_FIGO, Histologic_type, FIGO_stage, BMI, Age, Race, Ethnicity, Gender, Tumor_Focality, Tumor_Size_cm.

As a response, the agent shows a success message and a table preview. To produce this result, the agent automatically (1) generates Python code using the Pandas library that loads and selects the requested columns and (2) executes the generated code to load the CSV file into memory. This example illustrates the capability of *on-demand code generation* to accomplish a requested task.

Next, the user requests the system to perform schema matching between the table loaded and the attributes from the GDC standard:

> 👤 Match this to the GDC schema and fix any results that don't look correct.

As shown in Figure 2c, the system interacts with an LLM multiple times to answer this request. First, it sends the initial user's prompt to the LLM, which recognizes that the function match_schema is adequate to solve the request, and thus returns a tool calling instruction containing the match_schema function name along with the needed parameters. After receiving the instruction, the system executes the function, collects matches, and returns them to the LLM. Note that the LLM interprets the user's query and correctly identifies the function arguments: it uses gdc as the target schema and a default value for the method parameter in match_schema(). Alternatively, the agent also supports requesting a specific method in the prompt, offering experts greater control and customization.s

After analyzing the column match results received in the follow-up prompt, the LLM detects a mapping mistake for the column Histologic_type, which is incorrectly mapped to roots. To fix the error, the LLM triggers a tool calling instruction that uses the primitive function top_matches to find the top-10 alternative candidates for Histologic_type. After analyzing the results, it suggests using primary_diagnosis as a correction to the user.

```
🤖 The current mapping for 'Histologic_type' is 'roots', which seems incorrect.
The top alternatives are:
- history_of_tumor_type
- primary_diagnosis
- sample_type
I suggest using 'primary_diagnosis'. Do you agree with this choice?
```

This exemplifies an important benefit of the agent in augmenting and improving the effectiveness of the underlying libraries: using its knowledge, it is able to detect the error; then, it leverages the library to identify alternative choices. This also shows an example of complex task planning enabled by the agentic loop (Figure 2b).

The agent then displays a table of column mappings, where the "Source Column" contains columns from the source table (`dou.csv`), "Target Column" contains a column match from the GDC standard, and "Corrected?" indicates whether the LLM has corrected the original `bdi-kit` match.

```
🤖 Result:
Source Column            Target Column            Corrected?
[...]                    [...]                    [...]
Country                  country_of_birth         No
Histologic_Grade_FIGO    tumor_grade              No
Histologic_type          primary_diagnosis        Yes, from "roots"
Tumor_Size_cm            tumor_depth_measurement  No
```

Since the user is a domain expert familiar with the GDC standard, they ask for better matching options for `Tumor_Size_cm`. The system updates the previous list with a more appropriate match found using the `top_matches` function:

```
👤 Find alternative mappings for Tumor_Size_cm.
```

```
🤖 Result:
Source Column      Target Column                       Corrected?
[...]              [...]                               [...]
Tumor_Focality     tumor_focality                      No
Tumor_Size_cm      tumor_largest_dimension_diameter    Yes, from "tumor_depth...
```

After finding correct matches for all source columns, the next step is to find value mappings for each column match. The following prompt requests the agent to find such mappings:

```
👤 Find value mappings for the columns Country, Histologic_Grade_FIGO,
Histologic_type, FIGO_stage, Race, Ethnicity, Gender, Tumor_Focality. If
there are any errors in the mappings, please provide suggestions.
```

As in the previous prompts, finding value mappings requires multiple LLM interactions. First, the LLM call the `match_values()` function for each of the source-target column pairs using the default method `tfidf` (an efficient algorithm that chooses matches based on the tf-idf similarity over n-grams). After receiving the value mappings, the LLM verifies if they are correct and fixes the erroneous entries. To do so, it calls another function to discover the list of acceptable values in a GDC attribute and picks the correct one. A snippet of the result is shown next (we omit part of the output due to lack of space):

```
🤖 [...]

Histologic_Grade_FIGO → tumor_grade

Source Value    Target Value    Corrected?
FIGO grade 1    G1              Yes, from "Low Grade"
FIGO grade 2    G2              Yes, from "Low Grade"
FIGO grade 3    G3              Yes, from "Low Grade"

[...]

FIGO_stage → figo_stage

Source Value    Target Value    Corrected?
IIIB            Stage IIIB      No
IIIA            Stage IIIA      No
II              Stage II        Yes, from "Stage III"
```

Note that the LLM detects and corrects errors automatically without user intervention. For example, it corrects the match ⟨II, Stage III⟩ to ⟨II, Stage II⟩. This shows how we can leverage efficient algorithms to perform most of the work while resorting to LLMs as evaluators that can identify and fix mistakes only when needed.

The final step is to generate a harmonized table:

```
👤 Please create a final harmonized table based on the discovered column and
value mappings and save it at "dou_harmonized.csv".
```

```
🤖 The harmonized table has been successfully created and saved as
'dou_harmonized.csv'.
```

Here, the agent uses `materialize_mapping` to generate a table and save file 'dou_harmonized.csv'. The LLM first creates a declarative mapping specification according to the `bdi-kit`'s format used to describe a harmonization plan. Figure 4 shows a snippet of the generated harmonization plan, which includes a declarative specification of the mappings in JSON. In this example, we only show dictionary-based transformations that map source values into target values. However, `bdi-kit` also supports other types of transformations, such as custom mappings that take as input a custom Python function (or lambda). This can potentially be used along with the on-demand code generation.

The main advantage of using a declarative language to describe harmonization plans is that it enables reproducibility: once a plan is created, users can feed the plan along with the source data into `materialize_mapping` function to recreate the harmonized data. This does not require re-running any LLM-based interactions, since all transformations are encoded in the harmonization plan.

```
[
 [...]
 { "source": "Histologic_Grade_FIGO",
   "target": "tumor_grade",
   "matches": [ ["FIGO grade 1", "G1"],
                ["FIGO grade 2", "G2"],
                ["FIGO grade 3", "G3"] ] },
 [...]
]
```

**Figure 4: A snippet from the mapping specification generated that is passed to `materialize_mapping()` function.**

**Evaluation.** We conducted a preliminary evaluation and compared `Harmonia` against baseline methods from `bdi-kit` executed without agent support. The results, summarized in Table 1, show that `Harmonia` achieved the best performance across both tasks. For schema matching, `Harmonia` successfully associated the column 'Histologic_type' with 'primary_diagnosis', which contains 2,625 unique values in the GDC. For value mapping, `Harmonia` was able to correct values such as 'FIGO grade 1' to 'G1'.

**Table 1: Performance comparison for schema matching and value mapping tasks.**

| Task | Method | Accuracy | Precision | Recall | F1 |
|------|--------|----------|-----------|--------|-----|
| Schema Matching | Baseline | 0.88 | 0.78 | 0.78 | 0.78 |
| | Harmonia | **1.00** | **1.00** | **1.00** | **1.00** |
| Value Mapping | Baseline | 0.58 | 0.58 | 0.59 | 0.57 |
| | Harmonia | **0.68** | **0.69** | **0.69** | **0.68** |

# 5 RESEARCH OPPORTUNITIES

`Harmonia` shows the potential of LLM-based agents to orchestrate actions, evaluate function outputs, detect errors, and generate additional functions. However, there are still open research questions to expand the system's capabilities and improve its effectiveness and usability, which we outline below.

**Agent Evaluation & Benchmarks.** Most existing evaluation benchmarks are focused on isolated tasks, such as schema matching or entity linking [38, 44, 65]. However, agentic systems create a need for end-to-end evaluation benchmarks and metrics to measure progress effectively. Recently, researchers have started developing benchmarks for evaluating agents in various tasks, including data analysis and ML engineering [8, 27, 29, 76]. To spur and measure the progress in data harmonization, we need to create benchmarks tailored for this task.

**Data Integration Primitives.** *Uncertainty* quantification and *explanations* should be exposed by the primitives to guide decision-making [16]. Uncertainty in data integration arises from factors like ambiguous schema mappings and data values [68]. `Harmonia` tackles this by exposing similarity scores through its primitives. For example, a value matcher can return similarity scores so that the agent can trigger complementary primitives (e.g., value mapping) for deeper analysis. A key challenge is conveying the meaning of uncertainty measures from diverse primitives to LLMs and end users and instructing LLMs on how to use them [1].

Also, LLMs often lack transparency [30], so primitives must provide *interpretable explanations* to promote user trust. Primitives should offer clear usage documentation and expose their decision rationale. For instance, a matching algorithm description could document whether its similarity scores derive from syntactic similarity, semantic embeddings, or value distribution analysis. LLMs can also explain their decisions based on domain knowledge and primitive instructions, helping users better understand why a particular path was chosen [61, 77]. A key opportunity is training agents to discern when to rely on LLM explanations, apply alternative strategies, or engage with the user directly.

Mapping data between schemata involves resolving entities and transforming data [36]. LLM-based methods have proven helpful in entity resolution [20, 51], generating or finding transformations functions [10, 11, 15, 42, 53, 62], and evaluating LLM performance on such tasks [48]. However, integrating these methods into agent-based systems requires consistency across diverse data models and alignment with broader agent goals. Also, we need methods that allow agents to identify and recommend appropriate attribute transformations and suitable functions for a given input dataset. This is especially in challenging cases such as table restructuring or non-standard formats [42].

**Robustness and Reliability.** LLMs have shown inconsistency across various scenarios (e.g., as text summarization evaluators [60]), often producing varying results when executed multiple times [3]. This variability can undermine reproducibility and reliability, particularly in critical applications where consistent mappings or transformations are crucial. In our experiments, we observed that while the LLM typically identified and fixed incorrect mappings, it occasionally failed to do so (even when provided with the same prompts).

Handling large and complex tables with many attributes poses additional challenges, as these can lead to long chat histories that exceed the LLM context window. When this occurs, the LLM may lose access to earlier relevant information, thereby affecting the robustness. While approaches have been proposed to mitigate context window limitations (e.g., [32] [47]), it is not clear if they address the issues in agent systems. Equipping agents with access to read and store data in external databases (such as the Provenance DB discussed in Section 3) may be an effective solution to this issue.

**User-Agent Interaction and Interfaces.** To improve usability, agentic systems must go beyond natural language (NL) interfaces. While NL is flexible, it is also often ambiguous and may lead to under-specified task descriptions [75]. Since the same task can be expressed in multiple unpredictable ways, a mismatch between the user task descriptions and agent prompt specifications may occur. Therefore, detecting when clarifications are needed may help increase overall success [75]. These issues could also be potentially addressed by action-oriented UIs that recommend actions linked to predefined prompts. Moreover, using rich visual representations may be more effective at conveying information to the user.

**Provenance-Aware Agents.** Provenance-enabled systems have demonstrated promising results in data science pipelines [9, 54]. In data harmonization pipelines, we can track all interactions that contribute to obtaining a specific value mapping. For example, we could record all user-agent and agent-primitive interactions involved in determining the mapping of "`FIGO grade 1`" to "`G1`" (see Figure 4). This would allow tracing the lineage of all values in the output data. Moreover, this information could potentially be used to learn user preferences that reduce the need for user interactions [37]. By learning from provenance and pipelines accumulated over time, a system could further streamline the harmonization process by automating all steps and presenting final results directly to users.

**Data Harmonization Pipelines.** In our system, data harmonization is expressed as a pipeline where multiple primitives (predefined, user-defined, or agent-defined) are interconnected through their inputs and outputs to produce the final harmonized dataset. The pipeline can have various objectives, such as maximizing the number of correct column matches and value matches, minimizing the number of interactions with users, or minimizing the computational costs (e.g., runtime or LLM calls). Achieving these objectives represents an optimization problem, requiring the system to navigate a complex search space and balance multiple objectives to determine an optimal sequence of operations. This is similar to the process used by AutoML systems that automatically synthesize end-to-end pipelines [46]. One challenge for harmonization agents is that to guide the search, we need to design optimizers that measure harmonization success, a non-trivial tasks, and balance multiple objectives including computational costs.

**Acknowledgments.** We thank Brandon Rose and Jataware for supporting the implementation of Harmonia's initial prototype. This work was supported by NSF awards IIS-2106888 and OAC-2411221, and the DARPA ASKEM program Agreement No. HR0011262087 and the ARPA-H BDF program. The views, opinions, and findings expressed are those of the authors and should not be interpreted as representing the official views or policies of the DARPA, ARPA-H, the U.S. Government, or NSF.

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
