# OpenReview forum: "Interactive Data Harmonization with LLM Agents: Opportunities and Challenges"
_ACM.org/SIGMOD/2025/Workshop/NOVAS — NOVAS 2025_

### Official Review · Reviewer_Tx26 · 2025-04-19

**Confidence:** 4

**Improvement Opportunities:**

- The main drawback in this article is the lack of any quantitative evaluation. While the motivation leading to the proposal of the prototype is pretty well written, without any baseline evaluation we don't know how well this approach may work. I understand that there may not be a lot of work in this specific niche application. Maybe a non-agent based system could work as a baseline to build a quantitative evaluation.
- More information about the application of the prototype to the clinical data domain can also help to illustrate the usefulness of the approach. An error analysis over the application of the agent of the data can shed light on what type of issues arise when using the agentic loop for harmonization
- Given the level of detailing in the description and the components of harmonia, the system prototype is implied to be complete. Since the user interface is argued as one of the core components of the prototype, one or more screenshots, or a link to a video would strengthen a lot the prototype.

Addressing the points above will make a much stronger submission.

**Minor Comments:**

The _tombstone_ symbol at the end of the examples is a bit unusual, given that this isn't a mathematical proof.

**Short Summary:**

In this paper, the authors propose an LLM agent-based pipeline for data harmonization, _Harmonia_. The system is built upon a collection of data harmonization primitives and it is driven by an agent loop that selects which of those primitives to use to map a source table to a target table based on a data vocabulary and receiving feedback from the user through an interface when necessary.

One of the key observations is that LLMs contain general and domain specific knowledge in their parametric memory that can be leveraged by the harmonization process. This allows reducing the burden of tasks such as schema matching, entity linking, etc.

The proposed system is described using an example to map clinical data using a domain specific vocabulary (GDC).

**Strong Points:**

- The authors did a great job at narrating the motivation, describing the abstract harmonization pipeline and justifying how this problem is a good match for an agent-based system solution
- The system prototype and use case is pretty well described too, making really simple to understand it using examples provided as figures and diagrams that illustrate the execution flow of the agent
- Next steps are clearly identified and addressed for future work
- The proposed system makes use of a existing primitive library and states the underlying software packages used for its implementation

---

### Official Review · Reviewer_Croi · 2025-04-21

**Confidence:** 3

**Improvement Opportunities:**

I1: Allowing users to directly edit outputs could improve usability. It can be used as signals for provenance-aware agents to learn user preferences.

**Minor Comments:**

-

**Short Summary:**

This paper introduces Harmonia, an agentic system for interactive data harmonization using llms. It solves the problem of harmonizing multiple tables with inconsistent schemas and data. The authors built a chatbot prototype on top of a toolkit of composable data harmonization primitives and demonstrated its application on clinical datasets. The paper is well-written, easy to follow, and supported by clear examples. It also discusses key challenges and outlines promising research directions for future work.

**Strong Points:**

S1: Tackles a relevant problem. It is well motivated. Building agents for harmonization is a timely application of LLMs.

S2: The demonstration of the chatbot prototype is clearly described. It is easy to understand how the system works and how users can interact with it.

S3: The paper outlines several promising avenues for future work.

---

### Official Review · Reviewer_zX3d · 2025-04-25

**Confidence:** 5

**Improvement Opportunities:**

O1 The paper is framed as a vision paper but the ideas on data harmonization methods, system functionalities (i.e. beyond schema-, and value matching), and user-interactions are quite straightforward for LLM-powered data integration. Moreover, the paper does not present very new or specific challenges. Altogether, the "usefulness" of the paper is quite low.

O2 The steps of a data integration use-case are explained, presenting intermediate steps (user requests) along with outputs of the system,  but no technical details are provided about how the system is implemented: how are user inputs interpreted and refined, are there any data transformation steps executed, which tools are integrated, what models are used under the hood and how are they adapted to be effective on these tasks, etc?

O3 The paper discusses a prototype but does not present any (experimental) insights from it. If there are no experimental results yet, then a small user-study with the prototype would be nice to understand if the implemented functionalities and user interactions are effective.

O4 There is a large body of work around data integration and data wrangling, also using LLMs, which is not reflected or built on.

**Minor Comments:**

M1 The level of agency of the individual components is unclear: what capabilities do the agents have, and how do they interact with each other? It seems to be a fixed pipeline of LLM calls.

**Short Summary:**

The paper discusses the vision of LLM-powered interactive pipelines for tabular data harmonization. Example interactions and intermediate outputs of such a system are outlined, and research challenges are described. While the system sounds promising, the paper lacks: 1) specification of the system (and particularly the LLM-powered components), 2) experimental results or usability insights, and 3) novel and specific challenges.

**Strong Points:**

S1 The paper presents a vision of an LLM-powered data harmonization system, which is a relevant and promising research direction. It would serve as an interesting topic for discussion.

S2 The data integration process is well illustrated through a concrete use-case, and the paper could serve as a reference guide for such LLM-powered data harmonization systems.

S3 The paper presents research directions that can be addressed, with most detail for the Data Integration functionalities.